METHODS

# A neural network model enables worm tracking in challenging conditions and increases signal-to-noise ratio in phenotypic screens

Weheliye H. Weheliye[1,2], Javier Rodriguez[1,2], Luigi Feriani [1,2¤a], Avelino Javer[1,2¤b], Virginie Uhlmann[3], André E. X. Brown [1,2]*

1 MRC Laboratory of Medical Sciences, London, United Kingdom, 2 Institute of Clinical Sciences, Imperial College London, London, United Kingdom, 3 European Bioinformatics Institute (EMBL-EBI), Wellcome Genome Campus, Cambridge, United Kingdom

¤a Current address: Australian Synchrotron - ANSTO, Melbourne, Australia
¤b Current address: Apheris, Berlin, Germany
* andre.brown@imperial.ac.uk

## Abstract

High-resolution posture tracking of *C. elegans* has applications in genetics, neuroscience, and drug screening. While classic methods can reliably track isolated worms on uniform backgrounds, they fail when worms overlap, coil, or move in complex environments. Model-based tracking and deep learning approaches have addressed these issues to an extent, but there is still significant room for improvement in tracking crawling worms. Here we train a version of the DeepTangle algorithm developed for swimming worms using a combination of data derived from Tierpsy tracker and hand-annotated data for more difficult cases. DeepTangleCrawl (DTC) outperforms existing methods, reducing failure rates and producing more continuous, gap-free worm trajectories that are less likely to be interrupted by collisions between worms or self-intersecting postures (coils). We show that DTC enables the analysis of previously inaccessible behaviours and increases the signal-to-noise ratio in phenotypic screens, even for data that was specifically collected to be compatible with legacy trackers including low worm density and thin bacterial lawns. DTC broadens the applicability of high-throughput worm imaging to more complex behaviours that involve worm-worm interactions and more naturalistic environments including thicker bacterial lawns.

## Author summary

Measuring how animals move in videos is useful in genetics and neuroscience experiments. Humans are good at following moving animals, but computers struggled until around ten years ago when deep neural networks provided a way to recognise objects, including animals, if they were trained on many

**Data availability statement:** Data, code, and models are available from Zenodo: https://zeno-do.org/records/15526615.

**Funding:** This work was supported by a grant from the Medical Research Council (https://www.ukri.org/councils/mrc/) to AEXB (MC-A658-5TY30). The funders did not play a role in the study design, data collection and analysis, decision to publish, or preparation of the manuscript.

**Competing interests:** The authors have declared that no competing interests exist.

examples. Ironically, these methods have not always worked as well for the simplest animals like nematode worms because they don't have clear keypoints on their bodies like joints that the networks can learn to recognise. In this paper we show that a recently developed network that was designed specifically for slender objects like worms can be trained to recognise and track worms crawling on agar plates (a common lab environment) even when they are in thick food or overlapping with each other. This network uses information from a series of frames, instead of single images, to resolve difficult cases. Better tracking makes it easier to detect differences between worms treated with different chemicals which will improve future drug screens.

## Introduction

High-resolution tracking that measures both position and posture has long been applied to the nematode *C. elegans* [1–12] and has enabled applications in genetics, neuroscience, and drug screening among others. One reason that tracking postural details in *C. elegans* has been accessible since the turn of the century is that worms have a simple body morphology and can be imaged with high contrast while confined to two dimensions on the surface of an agar plate. High contrast imaging of sparsely distributed animals with a simple morphology is well suited to classic computer vision approaches for segmentation and skeletonization. Despite these successes in worm tracking, most methods developed for pose estimation (identifying posture and orientation) were brittle and simple overlaps between animals or self-intersection during coiling remained challenging for longer. Alternative methods were developed for pose estimation in overlapping worms [13,14] or individual coiling worms [15,16] but most of these methods are computationally intensive and less appropriate for high-throughput screening with large numbers of worms.

More recently, advances in deep learning have extended the applicability of pose estimation to more challenging applications including non-uniform backgrounds and animals with complex body plans [17–19]. Several instance segmentation methods have been applied to or developed for segmenting worms specifically [20–22]. A model called DeepTangle was recently developed to track swimming worms which takes into account temporal information and explicitly learns a worm shape model to produce skeletons directly from videos [23]. Swimming worms adopt simpler lower-dimensional postures than crawling worms, rarely coil, and tend to have simpler overlaps. Here we train DeepTangle on a new dataset including the more complex postures and long overlaps of crawling worms and show that the resulting model, DeepTangleCrawl (DTC), can track coiling and overlapping worms, achieving a new state-of-the-art on challenging worm tracking data compared to two instance segmentation approaches. DTC produces segmentations in more cases than the alternative models and the resulting segmentations are only somewhat less accurate in terms of root mean square deviation. Using DTC for tracking, we find that it produces tracks that are longer and more complete (fewer gaps) than those produced by

Tierpsy. The improved tracking enables the analysis of behaviour in previously inaccessible conditions and improves the signal-to-noise ratio in phenotypic screens.

## Results

### Data and model training

All the data we used for training and evaluation were collected using megapixel camera arrays with a resolution of 12.4 μm/pixel and a frame rate of 25 frames/second [24]. We selected subsets of data from diverse experiments including data from disease models [25], worms treated with pharmaceuticals [24], and worms treated with insecticides and nematicides [26]. The data were collected by different operators over several years. The goal was to have a training dataset that included variation in worm behaviour and morphology as well as technical variation, day-to-day variation, and seasonal variation. The current training data does not include data from different imaging setups or different labs.

The original DeepTangle model was trained on synthetic data. Simulating swimming worms is simpler than crawling worms for biological and technical reasons. First, worm behaviour during swimming is simpler than during crawling [5,27,28]. Second, imaging worms in liquid tends to result in a more uniform background than worms crawling on the surface of an agar plate seeded with a lawn of bacteria. Third, the challenging tracking examples we are most interested in solving—such as worms in thick bacterial lawns or tightly overlapping worms with correlated locomotion and an interface that is complicated optically—are the cases where producing reliable synthetic images would be the most challenging. Given these considerations, we constructed our training data from recordings of actual worms.

DeepTangle uses information from adjacent frames to improve segmentation. Therefore, all annotations were done on short clips consisting of 11 frames with all worms in the middle three frames annotated (Fig 1A). All 11 frames from each clip are input to the model during training. We first compiled an extensive training data set consisting of non-coiling non-overlapping worms that were accurately tracked using our baseline algorithm Tierpsy (Fig 1B). We then created a synthetic dataset by overlaying pre-segmented and skeletonized worms with random orientations to create training data with overlaps. These simple cases and synthetic overlaps were then supplemented with manual annotations from more challenging cases including overlapping worms, worms in shadows at the edge of plates, worms in thick food lawns, and coiling worms (Fig 1C). Most instance segmentation methods require only single annotated frames. To train these models we simply used the annotated frames individually.

Compared to the original DeepTangle model [23], we increased the dimension of the latent space representing worm shapes from 12 to 72. Because crawling behaviour has diverse timescales ranging from fast locomotion to long periods of dwelling, the 11-frame clips in the training data were uniformly sampled from 2, 4, and 8 second recordings so that the model could learn postural changes across several time scales.

Before both training and inference, we subtracted the background from recordings using singular value decomposition. Specifically, we performed SVD on videos that were subsampled 1:400 in time and used the single highest energy mode as a background image that was then subtracted from all video frames. This first mode consistently corresponds to the background without any worms. This step made all of the data, including the synthetic overlapping worm clips, directly comparable without a background outside of the worms.

### Pose estimation accuracy

We evaluated DTC on a set of held out data that were sampled across the same experiments as the training data but that were not seen by any model during training. The median root mean squared deviation (RMSD) between the predicted pose and the manually annotated pose (or Tierpsy pose in the case of simple non-overlapping shapes) is 2.2 pixels (Fig 2A). At the resolution used here, this corresponds to an RMSD of approximately half a worm body width. To compare the performance of DTC to the current state of the art, we used the same data to train Omnipose and a landmark-based tracker similar to SLEAP [29] we refer to here as PAF for part affinity field [30]. PAF and Omnipose have a lower modal

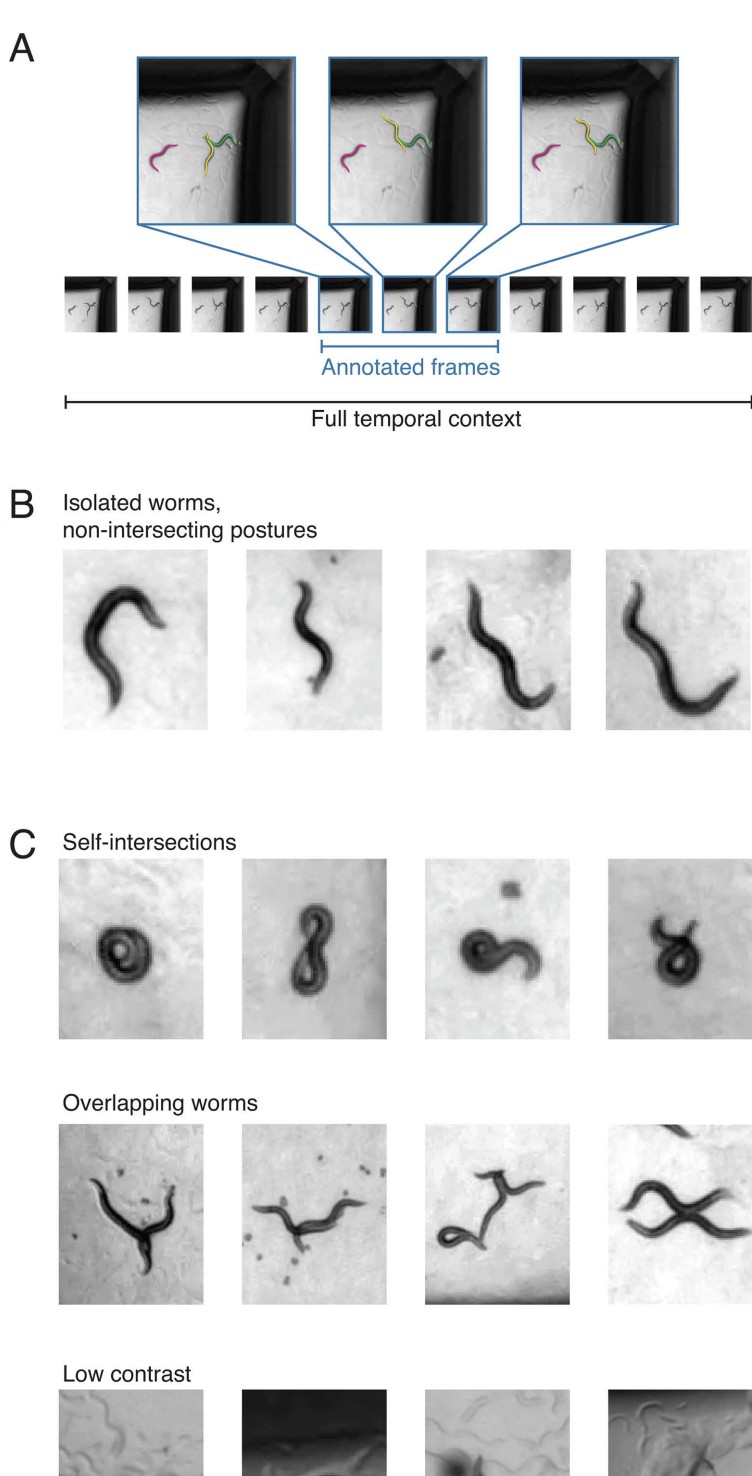

**Fig 1. Training data is a mix of straightforward and challenging cases. (A)** DeepTangle's input consists of clips with central frames annotated. Other models were simply trained on individual annotated frames. **(B)** Isolated non-intersecting worms can be tracked using Tierpsy's existing algorithm. **(C)** There are several categories of more challenging cases where simple skeletonization algorithms fail including self-intersecting worms, multiple overlapping worms, and worms with complex non-uniform backgrounds. Clips for these cases were manually annotated.

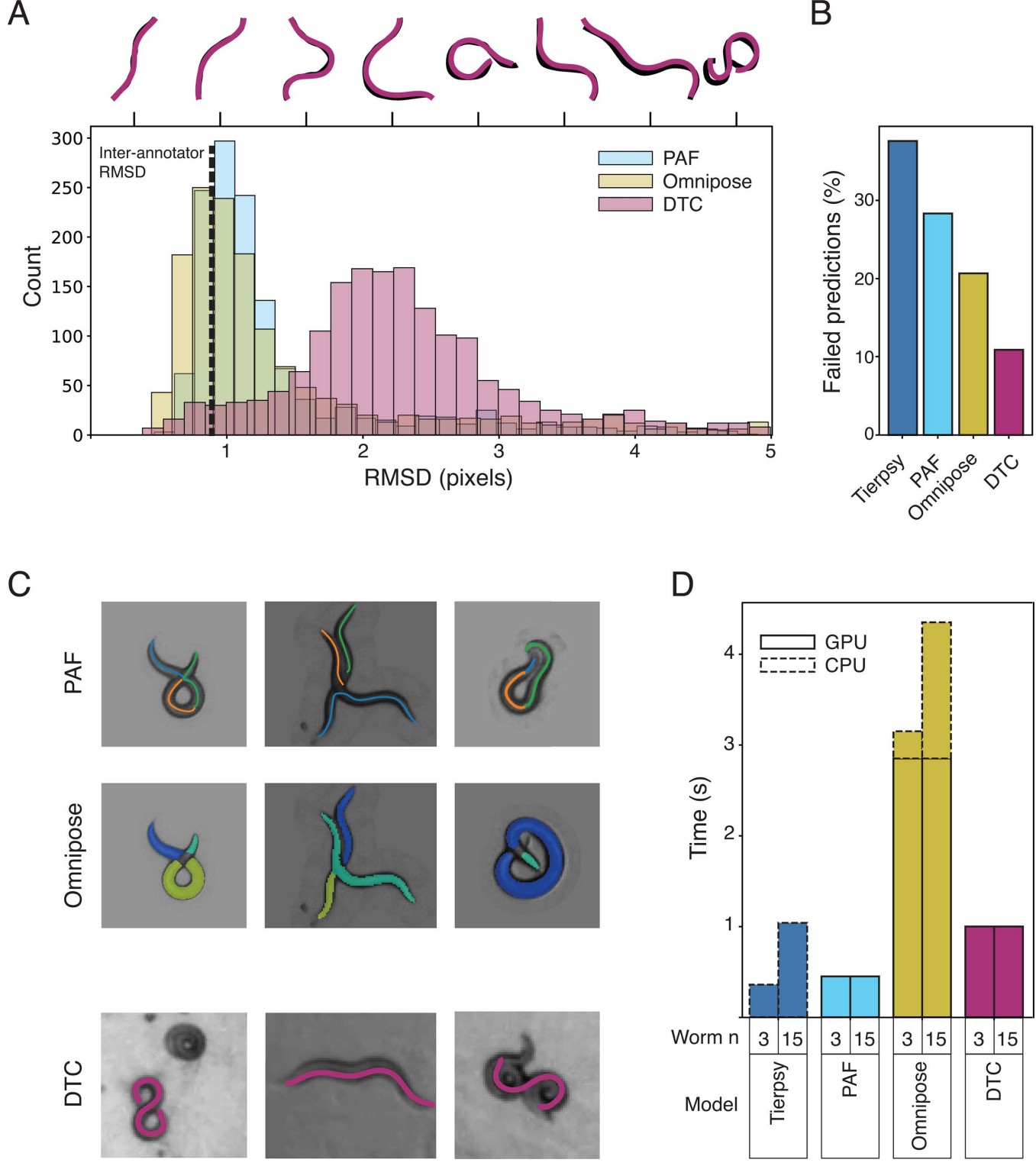

**Fig 2. Pose estimation accuracy. (A)** Histograms of root mean square deviation between held-out skeletons and predicted skeletons for each tested model. The vertical dashed line shows the RMSD between three manual annotators. Skeletons above the histogram are examples that illustrate the corresponding RMSD visually. **(B)** Bar chart showing the number of cases in the held-out test data where a model fails to make a prediction (e.g., Tierpsy

fails on coiled worms or a neural network model does not identify any worms above a confidence threshold). **(C)** Examples of the kinds of errors each model makes. PAF and Omnipose often over-segment worms. DTC fails when worms are fully coiled in circles or in tight parallel contact for an extended time. **(D)** Computation time per input frame for the different models as a function of worm number/well. Each camera records 16 wells in a 96-well plate so these correspond to 48 and 240 worms per video. Tierpsy only uses CPU computation while Omnipose uses GPU and CPU because we use Tierpsy's skeletonization algorithm to convert segmented regions to skeletons.

RMSD than DTC for the cases where a prediction is made. This difference persists when the models are compared separately to Tierpsy or manual annotations, although on the more difficult manually annotated cases PAF and Omnipose are more likely to show higher RMSD values (S1 Fig). The main source of this difference is accuracy around the beginning and end of the worms. A distance measure that focusses on midline accuracy shows less difference between the models (S1 Fig). Importantly, there are also cases where no prediction is made and this failure rate is different across models (Fig 2B). On this measure, DTC performs the best, making no predictions less frequently than other models.

Not only are the rates of failure different, the cases each model fails on are different. For example, Omnipose takes worm width as a parameter and is good at separating parallel worms that are in contact along their length but tends to over-segment coiled worms or crossing worms (Fig 2C, top row). PAF makes some similar errors on coiled and crossing worms and struggles to separate parallel worms (Fig 2C, middle row). DTC performs better on coiled and crossing worms, most likely because it is able to take earlier and later frames into account to resolve the most likely poses. DTC does still fail on worms that form long-lasting overlaps or tight coils with little motion (Fig 2C, bottom row). Because the worms do not move, DTC is unable to take advantage of temporal information to resolve the coiled shape. When these slow-moving coilers do straighten within a clip available to DTC, it is often able to estimate the pose in nearby coiling frames although it still fails for the tightest circular coils. All models currently fail in cases of complex overlaps with multiple worms (Fig 2C, bottom row, right).

In addition to their accuracy, computation time is an important metric to consider for any model that will be used for large scale phenotypic screens. We compared the inference time for each model separated into CPU and GPU time (Fig 2D). We only included the time to generate skeletons, not the time to connect them into tracks because the tracking algorithm could be varied and is not a core feature of each approach. The baseline model Tierpsy is run only on CPUs while PAF and DTC run only on GPUs. Omnipose outputs a mask for each worm and so requires a second skeletonization step. Here we apply Tierpsy's skeletonization algorithm and do this step on CPUs. The neural network methods process all pixels and so processing time does not depend on the number of worms. Tierpsy skeletonizes each segmented object separately and so processing time increases with worm number. To speed up inference in cases where there are a small number of worms, we down-sampled videos 1:3 in time and then used a three-dimensional smoothing spline to interpolate the missing data. This reduced total computation time with only a small decrease in performance, but the time advantage is modest with 15 worms per well (S2 Fig).

Given its state-of-the-art performance in number of worms skeletonized and its acceptable accuracy and computation time, we focus on characterising DTC for the remainder of the paper. The results below were calculated using the DTC+spline algorithm but the results are broadly similar with DTC applied to each frame.

## DTC improves tracking in challenging conditions

The data presented so far were collected in conditions with relatively sparse worms on thin food lawns that were optimised for tracking using Tierpsy. Tierpsy is more sensitive than DTC to thicker food lawns, the presence of eggs or other background objects, as well as more frequent overlaps or coiling. The difference in performance in more challenging conditions is therefore larger. When there is a higher density of worms and eggs on the lawn, DTC is still able to skeletonize the majority of worms accurately (Fig 3A and S1 Movie) whereas Tierpsy frequently fails (S2 Movie). Because skeletonization is maintained for overlapping worms, DTC is also able to accurately maintain worm identity through collisions (Fig 3B).

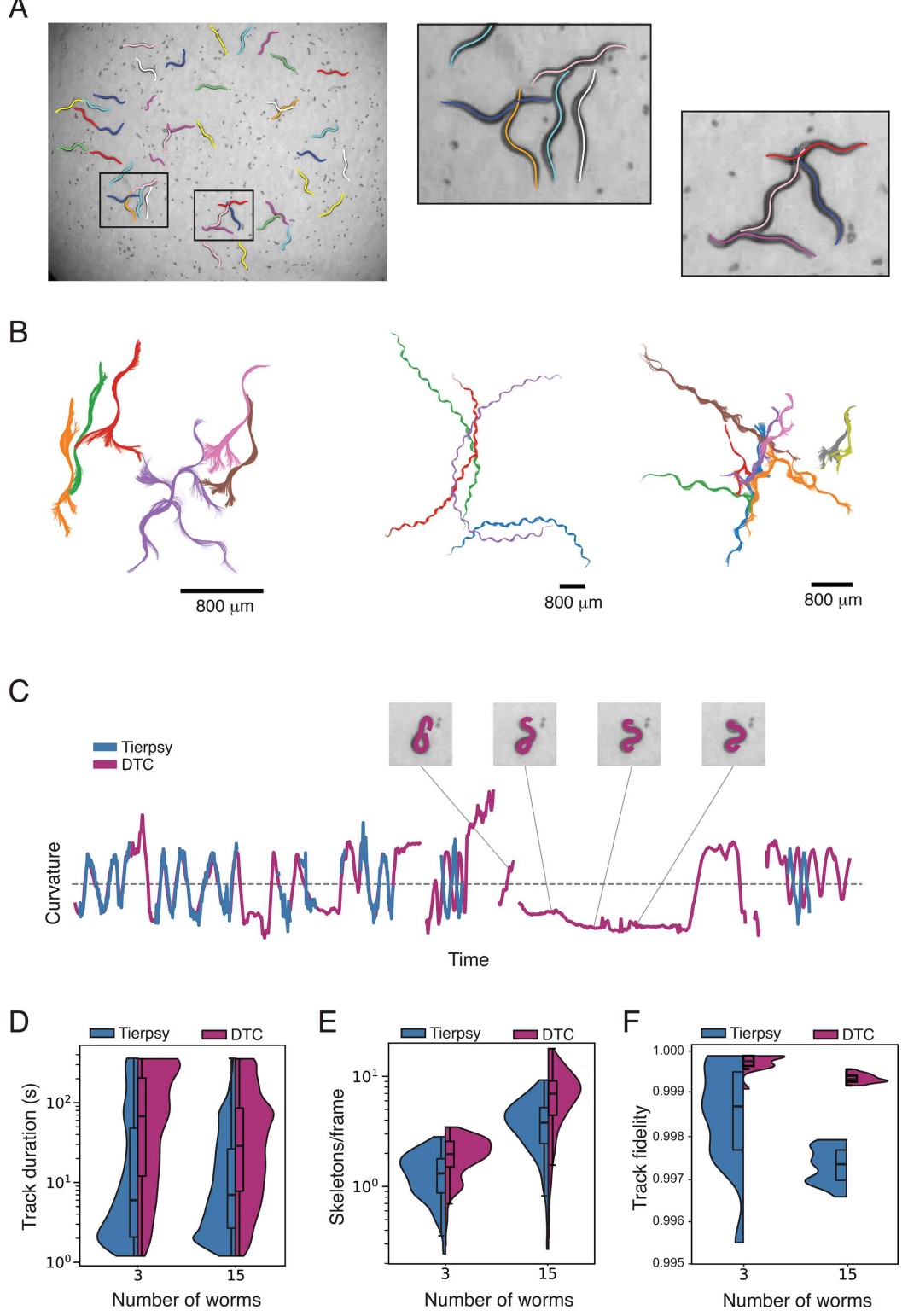

**Fig 3. Tracking in challenging conditions. (A)** Multiple overlapping worms with high density of eggs. Inset images show higher magnification images of two sets of overlapping worms where all individuals are successfully skeletonized. **(B)** Examples of continuous tracks that preserve worm identity during and through collisions. **(C)** For highly curved worms that form long-lived coiling shapes, long gaps in the data can be present in Tierpsy-derived

data. Here, there is a long gap with a high curvature that is recovered using DTC. **(D)** Improved skeletonization leads to longer tracks from DTC compared to Tierpsy. Note the log scale. The duration is longer for videos with 3 and 15 worms per well. **(E)** The number of skeletons/frame averaged over a video. DTC tracking produces numbers closer to the nominal number of worms per well. **(F)** The fraction of objects that is correctly tracked across frames compared to manually corrected trajectory data for videos with 3 and 15 worms per well.

When worms perform sharp turns, their self-intersecting poses cannot be resolved by Tierpsy, leading to gaps in data. For strong coiler mutants or worms treated with drugs that cause coiling, these gaps can be long and the poses that are not resolved are precisely those that are most characteristic so missing them limits accurate phenotyping. DTC is often able to resolve these shapes if there is some worm motion (Fig 3C and S3 Movie) although tight coils remain challenging, as mentioned previously (Fig 2C). Quantitatively, we see an increase in both trajectory duration and the number of skeletons per frame between Tierpsy and DTC for both 3 and 15 worms per well (Fig 3D).

To confirm that the longer trajectories are also more accurate, we manually corrected tracks for 3 and 15 worm videos and used the corrected data as the ground truth to calculate the tracking fidelity (the fraction of objects in each time step whose identity is correctly maintained) for DTC and Tierpsy. DTC produces more accurate tracks than Tierpsy (Fig 3F). Finally, to determine whether the increased accuracy was due to improved segmentation or DTC's tracking algorithm, we used nearest-neighbour tracking on the segmentation output from DTC and find an intermediate accuracy showing that both improvements in segmentation and the tracking algorithm contribute to DTC's performance compared to Tierpsy (S3 Fig).

### Improved signal to noise ratio in phenotypic screens

To determine whether improved tracking quality leads to improved sensitivity in the context of a large phenotypic screen, we reanalysed a previously published dataset consisting of videos of worms treated with diverse insecticides and nematicides [26]. Using Tierpsy tracker, we have previously shown that we could detect the effects of insecticides and also predict their mode of action based only on their behavioural effects. As noted in the previous section, some of these treatments led to worms that adopted self-intersecting postures that were not tracked by Tierpsy leading to significant missing data. We extracted the same behavioural features from videos tracked using Tierpsy and DTC.

As expected for relatively clean data that was collected specifically to work with Tierpsy, the results are broadly comparable using both tracking methods. For example, the median midbody speed calculated using skeletons from Tierpsy is highly correlated to the speed calculated using skeletons from DTC (Fig 4A). Looking across the Tierpsy256 features, the modal correlation coefficient is 0.71 (Fig 4B, left). Some features show little correlation between the two tracking methods. Many of these poorly correlated features also have low F-statistic calculated across drug classes suggesting that they are simply noisy features in this dataset (Fig 4B, right).

We next focussed specifically on a spiroindoline drug (SY1713) that causes strong coiling where we expect a difference between Tierpsy and DTC. For each dose, we find a higher curvature estimate from DTC than for Tierpsy, consistent with the expectation that DTC skeletonizes more frames with coiling (Fig 3C). The distribution of curvature values is also narrower for DTC-calculated curvatures, corresponding to a larger signal-to-noise ratio comparing drug treatments to DMSO controls for feature data derived from DTC tracking. Based on this observation we quantified the effect size for all features and all samples using features derived from Tierpsy and DTC tracking (Fig 4D). Effect sizes are correlated but are larger (below the diagonal in Fig 4D) for DTC-derived features.

### Discussion

DTC tracks more worms in challenging conditions including complex backgrounds and overlaps than the other tested models. It also has acceptable accuracy, with an RMSD of around half a worm width, although the current version is less accurate than Omnipose and PAF for the cases where these models make any prediction. The higher RMSD results from two kinds of errors. First, points can be off-centre and not pass through the midline of the worm. Second, points can be

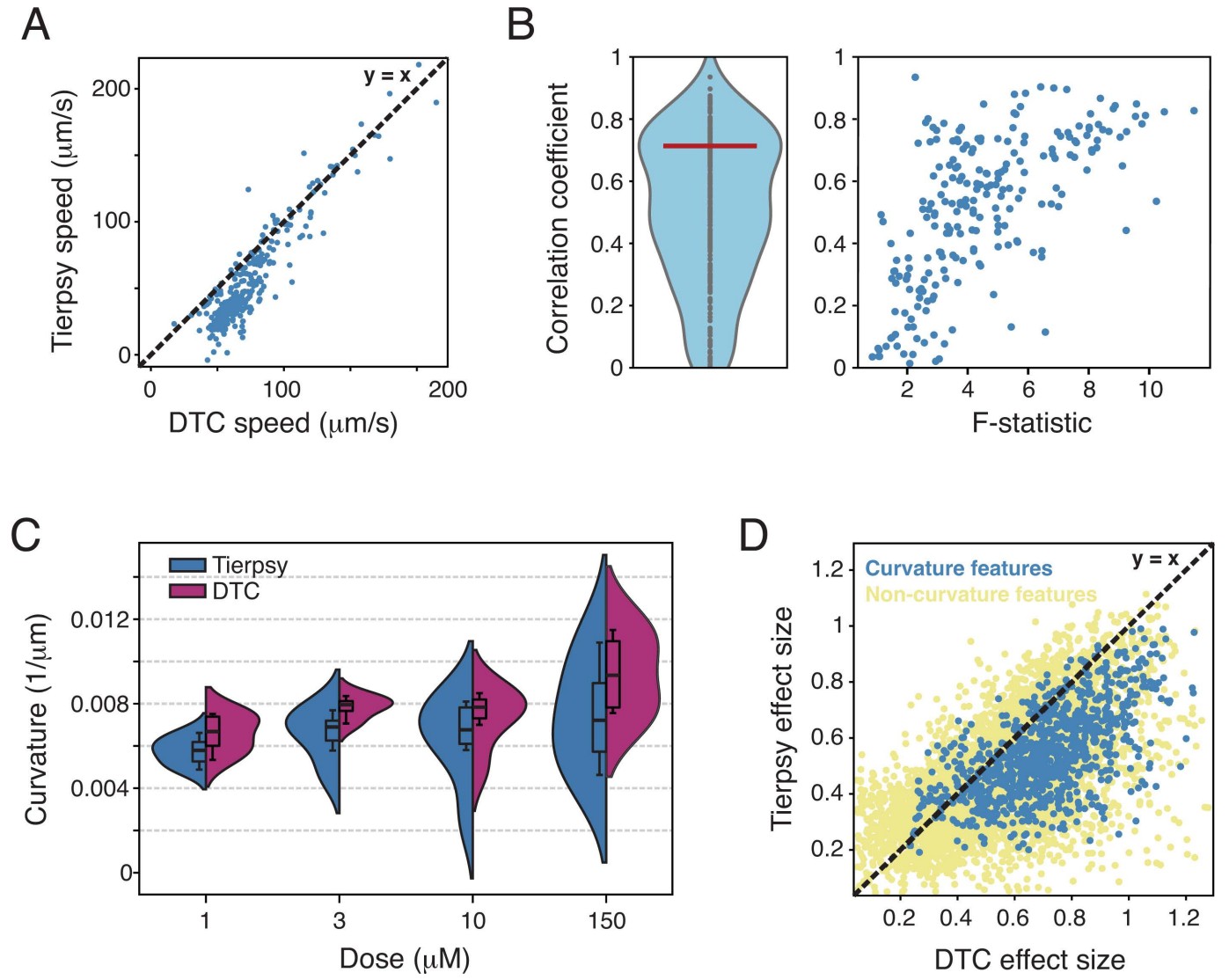

**Fig 4. DTC improves the signal to noise ratio in a phenotypic screen. All data from ref. [26]. (A)** Speed calculated from Tierpsy skeletons and from DTC skeletons for a random sample of 290 wells from a previously published drug screen. The dashed line is y = x. **(B)** Correlation coefficients for each of the Tierpsy 256 behavioural features for the data from ref. [26] (left). The red line indicates the modal value of 0.71. Correlation coefficient plotted against the F-statistic for each feature calculated over the entire dataset. **(C)** Tail curvature as a function of dose for worms treated with a spiroindoline known to cause coiling. **(D)** A comparison of the mean absolute value of Hedge's d effect size calculated using features derived from Tierpsy and DTC tracking data. Each point is the mean Hedge's d across all doses of a drug compared to DMSO controls for a single feature. Any feature with 'curvature' in the name except for time derivatives of curvature are shown in blue. All other features are shown in yellow.

on-centre but shifted along the length of the worm because the curve does not completely reach the head or tail of the worm. Even if two curves pass perfectly through the midline, there can be a substantial error measured by RMSD if one curve starts at the head and the other at the neck (S1 Fig). Some of these inaccuracies can be improved if the output of DTC or the original DeepTangle model is used as input to a differential rendering algorithm [31].

Because it captures worm shapes in more conditions, DTC opens the door to experiments that were previously difficult or impossible. For example, thicker bacterial lawns that are closer to natural worm habitats obscure worm outlines and

lead to more worm-like tracks that are difficult for some algorithms to distinguish from worms themselves. Perhaps more interestingly, a quantitative study of worm-worm interactions including their dependence on other behavioural variables like posture and speed is within reach. Although we do not expect to be able to track worms in large dense aggregates from brightfield imaging alone, studying the early stages of aggregate formation involving two- and three-worm interactions may already yield insight into aggregate formation. Because this analysis is compatible with large-scale imaging, it will be possible to look for mutants and natural variants that affect the early interactions and may reveal novel genes involved in worm behaviour.

Even in conditions optimized for classic segmentation and skeletonization algorithms, DTC increases the fraction of frames that yield reliable posture data, minimising gaps and preserving identity through overlaps and coils. This improvement directly translates into more data points per imaging run and increased effect sizes in phenotypic screens. The increased power comes from a better ability to detect subtle differences in behaviour as well as the ability to detect more extreme phenotypes such as the persistent high curvature seen in Fig 3C which is effectively never observed in control worms.

Despite these advances, DTC still has limitations. For example, the long-lived close interactions that are characteristic of mating remain a challenge for DTC. We expect that a large-scale study of mating would become possible with more mating-specific training data. Another potential improvement would be to use adaptive temporal down sampling at inference time so that the model has the best chance to use frame-to-frame differences to accurately identify worms. For example, worms that are nearly stationary could be tracked using clips generated from only every $100^{th}$ frame so that there is discernible motion to help resolve coils or complicated overlaps. An obvious downside of this kind of approach is the extra complexity required to determine the right degree of down sampling for each instance. More generally, a significant increase in training data will likely be required to make a model that generalises to data collected on other trackers. We have included code for an annotator that makes generating new training data relatively straightforward and would welcome a collaborative approach to worm annotation to generate multi-lab training data. A complementary approach would be to generate better fully synthetic training sets that capture complex behaviours like mating. Training data has been generated for single crawling worms in single frames and for multiple swimming worms in the original DeepTangle paper. It is possible to generate realistic worm crawling trajectories using generative models [32,33], however, these data are generated from existing behaviour data which is derived from legacy trackers. The motion models are therefore not trained to generate for interacting worms. Swimming worms interact weakly, but the behaviour of crawling worms is dependent on neighbours. For example, crawling worms can align parallel to each other and stay in contact with relatively small motions back and forth for extended periods. During contact, their postures are highly correlated. Further work is needed to accurately generate trajectories for strongly coupled crawling worms as well as to turn these trajectories into movies that are useful for training segmentation and tracking models.

Despite these limitations, DTC increases the value of data that is expensive to generate in terms of reagents (for drug screens) and labour for a modest investment of GPU time.

## Materials and methods

### Training data

Initially, isolated worm clips from Tierpsy were organized into sequences of 2, 4, or 8 seconds with the background subtracted. Each training clip consisted of a stack of 11 frames, each frame being 512x512 pixels. In total, we created 50,000 such clips. To simulate overlapping, we paired and stacked two clips on top of each other. Importantly, for each epoch, the dataset was randomly shuffled so that at every epoch, different pairs of clips were stacked together. This approach introduced variability and diversity to the training data. As a result, the effective number of unique clips used for training was 25,000 after stacking. The Tierpsy-derived data were supplemented with manual annotations for challenging cases

included coils, complex overlaps, and non-uniform backgrounds. A total of 1000 clips were annotated using a custom graphical user interface (GUI) (https://github.com/WeheliyeHashi/Worm_annotation). To create new manually annotated training data, we have also included a script that converts videos to clips that can be loaded and annotated using the GUI. For testing, we used 2000 skeletons, which are available on Zenodo (https://zenodo.org/records/15526615).

## DTC training and hyperparameters

Training code was adapted from ref. [23] and is available on GiHub (https://github.com/WeheliyeHashi/tierpsy_tracker_2.0). Training was conducted on an NVIDIA RTX A6000 with 48 GB of RAM. The model was implemented using JAX, with the Adam optimization algorithm, a learning rate of 0.001, and a modified ResNet (average pooling instead of max pooling in the first layer to break translational invariance) as the backbone architecture. The latent dimension was set to 8, and the maximum number of skeletons per grid was also set to 8. A cutoff distance of 48 pixels was applied during training, and the Principal Component Analysis dimension was set to 72.

The PCA transformation matrix was constructed using skeletons from both the training clips and supplemented with single-frame annotation data. As the model is based on a YOLO algorithm, it produces skeleton predictions, confidence scores, and latent space outputs. The output shape for the skeleton predictions was (32, 32, 8, 3, 72), corresponding to a 32x32 grid, with each cell measuring 16x16 pixels. Each cell contains 8 predicted skeletons across three central frames of the input stack, along with 72 PCA components. To convert these PCA components into real-space coordinates, we multiplied them by the PCA transformation matrix derived from the training data and single-frame annotations. The output shape for confidence scores was (32, 32, 8), and for latent space outputs, it was (32, 32, 8, 8). For inference, input images were padded so that the image resolution was a multiple of 16.

## Fitting 3D splines

For small numbers of worms, we found that it was faster and only slightly less accurate (S2 Fig) to down-sample videos 1:3 in time before applying DTC and then using a 3D smoothing spline to interpolate over missing frames [34]. The comparisons shown in Figs 3 and 4 were done using this down-sampling and interpolation procedure.

## Omnipose training and hyperparameters

The Omnipose code was adapted from ref. [22]. Training was conducted on an NVIDIA RTX A6000 GPU with 48 GB of memory. The model was implemented using PyTorch, with the Adam optimization algorithm, a learning rate of 0.1, and UNET as the backbone architecture. During training, we used the average cell diameter method as input, providing the estimated diameter for each worm. However, during inference, the actual worm diameter in pixels was used to enhance prediction accuracy. The training data, the trained model, and the inference code are available on Zenodo (https://zenodo.org/records/15526615).

## PAF training and hyperparameters

We implemented the 'Part Affinity Fields' (PAF) model, based on ref. [30]. The PAF model uses vector fields to represent connections between body parts. We adopted a bottom-up approach, where all body parts are detected simultaneously in the image, unlike top-down methods that first detect individuals and then estimate their poses. The model predicts the location of landmark points along the worm body and PAFs to determine the connections between them. To link two landmarks, the algorithm calculates a confidence score by integrating the PAF values along the line segment connecting the two points. Higher scores indicate stronger connections, allowing the model to accurately link the correct body parts and construct worm skeletons. This method considers both the spatial positions and the relationships between the body parts to ensure coherence. The implementation was carried out in PyTorch, using the Adam

optimizer with a learning rate of 0.001 and a UNET-based backbone architecture. To reduce under-segmentation of closely aligned worms, the spreads of both the part affinity fields (edges) and confidence maps were set to match the diameter of the worms.

**RMSD calculation and failed predictions**

To calculate the Root Mean Square Deviation (RMSD), we measure the minimum distances between the predicted skeleton points and the labelled points. Before calculating the RMSD, we ensure that the first point of the skeleton data corresponds to the head and the last point corresponds to the tail. This alignment is achieved using the head-tail algorithm in Tierpsy [12]. Points are interpolated to be equally spaced before calculating the RMSD.

‘Failures’ occur when no prediction is made. For Tierpsy, this decision is based on a series of heuristics. For example, the segmented area to be skeletonized must have exactly two points of curvature above a threshold (the head and tail) and the maximum and minimum width of the worm after skeletonization must not differ more than a preset threshold (helps to detect coiled shapes with an incorrect skeleton). For DTC, failure occurs when the model does not predict any skeleton above a confidence score of 0.5. For PAF, we check the number of detected landmark points. If a skeleton has fewer than 14 points (the expected number for a complete skeleton), we classify it as a half or broken skeleton and call it a failed prediction. For Omnipose, we adopt the same approach as Tierpsy since we use the Tierpsy skeletonization algorithm to go from segmented objects to skeletons.

To quantify tracking accuracy, we define the tracking fidelity

$$1 - \frac{\sum_t mme_t}{\sum_t g_t},$$

where $mme_t$ is the number of mismatches at time $t$ (from ID switches during tracking), $g_t$ is the number of ground truth objects at time $t$, and the sums are over all frames in a video.

**Phenotypic screen analysis**

We used the Pearson correlation coefficient to evaluate the linear association between the features obtained by Tierpsy and DTC. We used the F-statistic to quantify the signal in a given feature, calculated across all drugs and doses. Hedge's d was used to quantify the standardized difference in effect sizes between the two methods. Hedge's d is an adjusted version of Cohen's d designed to correct for small sample bias. These values were converted to absolute values, disregarding the direction of the effect, and then averaged across all drugs and doses for each feature.

**Supporting information**

**S1 Fig: Skeleton accuracy for test data subgroups and types of errors.** (A) Root mean square deviation between model predictions and test data for ‘easy’ cases which were skeletonized by Tierpsy. By construction the number of failed predictions for Tierpsy is zero. (B) The same as A but using manually annotated skeletons as the test data. These data have more difficult cases, but there are still a substantial number that Tierpsy is able to make predictions for (Tierpsy fails around 35% of the time on these data). (C) Examples showing manual annotations (blue) along with DTC predictions in red with different values of RMSD. In all cases, the predicted skeletons are close to the worm midline and the RMSD is driven mostly by overshooting at the head and/or tail. (D) Instead of measuring the point-to-point RMSD, we can measure the distance to the nearest portion of the test segment for each of the model predictions. (E) The mean distance computed as shown in D is more similar across the models although DTC still performs slightly worse.
(PNG)

**S2 Fig. Skipping frames and interpolating skeletons with a spline achieves similar accuracy and faster computational time.** (A) Histograms of root mean square deviation between held-out skeletons and predicted skeletons for each tested model, here including 'DTC + spline' in which only every third frame is tracked using DTC and skeletons for intermediate frames are interpolated using a 3D smoothing spline. The vertical dashed line shows the RMSD between three manual annotators. Skeletons above the histogram are examples that illustrate the corresponding RMSD visually. (B) Bar chart showing the number of cases in the held-out test data where a model fails to make a prediction (e.g., Tierpsy fails on coiled worms or a neural network model does not identify any worms above a confidence threshold). (C) Computation time per input frame for the different models as a function of worm number. Tierpsy only uses CPU computation while Omnipose uses GPU and CPU because we use Tierpsy's skeletonization algorithm to convert segmented regions to skeletons. DTC+spline uses both GPUs for predicting skeletons on the subsampled frames and CPUs for fitting smoothing splines to interpolate the missing frames. The advantage is modest for videos with 15 worms/well (which is nominally 240 worms/video since each camera records 16 wells on a 96 well plate).
(PNG)

**S3 Fig. Contribution of segmentation and tracking algorithm to DTC performance.**
(PNG)

**S1 Movie. DeepTangleCrawl skeletons (red) overlaid on a video with multiple overlapping worms and eggs in the background.**
(MP4)

**S2 Movie. Tierpsy skeletons (red) overlaid on a video with multiple overlapping worms and eggs in the background.**
(MP4)

**S3 Movie. DeepTangleCrawl skeleton (red) overlaid on a coiling worm.**
(MP4)

## Acknowledgments

We thank Bonnie Evans, Eleanor Warren, Riju Balachandran, and Tom O'Brien for help annotating data.

## Author contributions

**Conceptualization:** Weheliye H Weheliye, Avelino Javer, Virginie Uhlmann, André EX Brown.

**Data curation:** Weheliye H Weheliye, Javier Rodriguez, Luigi Feriani, Avelino Javer.

**Formal analysis:** Weheliye H Weheliye, Javier Rodriguez, Avelino Javer.

**Funding acquisition:** André EX Brown.

**Investigation:** Weheliye H Weheliye, Javier Rodriguez, Luigi Feriani, Avelino Javer, Virginie Uhlmann, André EX Brown.

**Methodology:** Weheliye H Weheliye, Avelino Javer, Virginie Uhlmann.

**Project administration:** André EX Brown.

**Resources:** André EX Brown.

**Software:** Weheliye H Weheliye, Avelino Javer, Virginie Uhlmann.

**Supervision:** Virginie Uhlmann, André EX Brown.

**Validation:** Weheliye H Weheliye, Javier Rodriguez, Luigi Feriani, Avelino Javer.

**Visualization:** Weheliye H Weheliye, Javier Rodriguez, Luigi Feriani, Avelino Javer, Virginie Uhlmann.

**Writing – original draft:** Weheliye H Weheliye, André EX Brown.

**Writing – review & editing:** André EX Brown.

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
