## [Decision Letter · Decision Letter 0]

11 Feb 2025

PCOMPBIOL-D-25-00022

An improved neural network model enables worm tracking in challenging conditions and increases signal-to-noise ratio in phenotypic screens

PLOS Computational Biology

Dear Dr. Brown,

Thank you for submitting your manuscript to PLOS Computational Biology. After careful consideration, we feel that it has merit but does not fully meet PLOS Computational Biology's publication criteria as it currently stands. Therefore, we invite you to submit a revised version of the manuscript that addresses the points raised during the review process.

Please submit your revised manuscript within 60 days Apr 13 2025 11:59PM. If you will need more time than this to complete your revisions, please reply to this message or contact the journal office at ploscompbiol@plos.org. Please include the following items when submitting your revised manuscript:

We look forward to receiving your revised manuscript.

Kind regards,

Adriana San Miguel

Academic Editor

PLOS Computational Biology

Tobias Bollenbach

Section Editor

PLOS Computational Biology

**Additional Editor Comments:**

Thank you for your submission, we have now received reviews which have raised some points to address. Please address the comments raised by the reviewers, such as additional clarification of data and methods, comparisons with other methods, etc.

**Journal Requirements:**

1) Please provide an Author Summary. This should appear in your manuscript between the Abstract (if applicable) and the Introduction, and should be 150-200 words long. The aim should be to make your findings accessible to a wide audience that includes both scientists and non-scientists. Sample summaries can be found on our website under Submission Guidelines:

3) We notice that your supplementary Figures are included in the manuscript file. Please remove them and upload them with the file type 'Supporting Information'. Please ensure that each Supporting Information file has a legend listed in the manuscript after the references list.

4) Please ensure that the funders and grant numbers match between the Financial Disclosure field and the Funding Information tab in your submission form. Note that the funders must be provided in the same order in both places as well.

**Reviewers' comments:**

Reviewer's Responses to Questions

**Comments to the Authors:**

Reviewer #1: The authors present significant progress in enabling the tracking of crawling nematodes in challenging environments. In my view, the results are novel and represent the current state-of-the-art in tracking. However, the presentation of the paper could be improved. I recommend that the authors carefully revise the manuscript to enhance its professionalism and clarity. Beyond that, I have specific comments that I hope the authors will consider:

- In order to understand RMSD, it is important to be able to understand your data. How much of this data was hand-annotated? If I understand your definition of "Inter-annotator RMSD" correctly, if all the data was hand-annotated, then all the ML models would have RMSD with mean around this value (~4.5 px), since the labels themselves would have this variance. Some details on the data and a discussion of this is warranted.

- The difference in accuracy between omnipose and DTC is interesting. I suppose this is due to omnipose working directly on pixels, but DTC predicting coordinates. Have you tried any type of post-processing refinement? For instance, the DTC output could initialize an "Active Contour Model" or similar that could adapt to the pixels. Could be discussed.

- You write "Because the worms do not move, DTC is unable to take advantage of temporal information to resolve the coiled shape.". Earlier you mentioned that you take the 11-frame window to mean different durations during training. Could you not do a similar trick during inference? i.e. somehow make the windows longer when there is less motion? Could be discussed at least.

- You mention original DT was trained with synthetic data. Was is the state-of-the-art of crawling worm simulations?

- Finally, maybe I misunderstood something, but I think the title, "An improved neural network model enables worm tracking in challenging conditions and increases signal-to-noise ratio in phenotypic screens", misrepresents the paper. As far as I can tell, the authors used a preexisting neural network, and did not make changes to this, but rather used a new type of data. If that is indeed the case, I would instead suggest "A neural network model enables worm tracking in challenging conditions and increases signal-to-noise ratio in phenotypic screens" for the title.

Minor:

Fig. 1A: The arrows confused me slightly. Could <---- be replaced with |------|? (if you agree)

Reviewer #2: The authors introduce DeepTangleCrawl method to infer the skeletons of overlapping C. elegans in images. The method is based on previously published DeepTangle approach that was developed on images of swimming worms. Here it was retrained on images of crawling C. elegans whose poses are more challenging to resolve. The paper is well written and clear I have however a few concerns about the types of comparisons the authors do to assess the method’s performance.

1. Both Omnipose and PAF were not developed specifically for C. elegans. Additionally, a keypoint-based method such as PAF is not well suited for resolving the pose of an organism without skeleton and limbs. I suggest that a C. elegans-specific segmentation and pose estimation method (some are cited in line 45) would be a more fair comparison of the performance.

2. line 52: “achieving a new state-of-the-art on challenging worm tracking data compared to two instance segmentation approaches.“ Segmentation and tracking are 2 different tasks and why would one compare segmentation methods in tracking performance? It is again not a fair comparison as tracking methods are optimized for tracking and segmentation methods for segmentation. Also, which instance segmentation methods the authors refer to?

3. I am also curious to know whether the method allows to segment tightly coiled worms, such as on Fig. 2c if they are not coiled persistently? Do the authors have an example where motion of the worm allows to disentangle the tightly coiled posture?

4. How is the tracking performed? There is no information on how the individual poses are linked into trajectories. Is it the same strategy as Tierpsy? In that case the tracking performance difference is due uniquely to the improved pose estimation.

5. Is there tracking groundtruth for these data? Can the authors present any relevant tracking metrics such as percentage of correctly tracked or the MOT metrics? Otherwise, I question whether the authors can indeed make a statement about the method’s tracking performance. There is simply more, better-resolved poses which allow to generate a higher number of longer trajectories, which are not necessarily correct.

6. There is a typo “identify” in caption of Fig. 3

Reviewer #3: The submitted article reports the development of an improved deep learning based worm tracker that delivers better signal to noise ratio and also carries segmentation and tracking on agar surfaces and thick lawns. The work, called DeepTangleCrawl (DTC), takes off from the earlier DeepTangle developed by A. Alonso and J. B. Kirkegaard Commun. Biol (2023). The manuscript is well written and presented. However, I have the following specific comments.

Specific comments:

1. On line 85: Specifically mention which challenging cases were annotated manually.

2. The work uses 11-frame clips for training purposes. Is this optimal? A comparison or sense of how many frames should be enough needs to be mentioned.

3. Line 88-89 mentions “Compared to the original DeepTangle model, we increased the dimension of the late space representing worm shapes from 12 to 72.” What is late space? And, it is not clear what 12 to 72 means. What units are these in?

4. As per Fig. 2A in section “Pose estimation accuracy”, RMSD is high for the DTC. Since RMSD computes the deviation in the minimum distances between the predicted and labelled splines. Intuitively, a better algorithm should reduce RMSD. What does an increased RMSD then mean?

5. DTC uses the same code as DeepTangle. It is not clear what additions or improvements are made to the original DeepTangle that lead to improved segmentation in crawling worms? This should be made explicit and highlighted appropriately.

6. Line 126 mentions that all models fail in cases of complex overlap. What could be a potential solution?

7. Video Movie_S1.mp4 shows the segmentation in a video, but also shows that there are frame drops. How often are the frame drops, which may lead to ID change or switch in worm ID? This, therefore, needs to be quantified and statistics should be reported.

8. Finally, a general comment I have is that the main achievement or improvement of DTC is better segmentation of worms in challenging poses, such as self overlapping, multiworm overlaps, coiling, etc. However, the term “pose estimation” used in the manuscript seems to be misleading, as on first reading, the impression was that it would be able to distinguish between poses such as omega turns, reversals, etc. However, the main point here is that it is merely able to carry out the detection of the worm even when they are in these poses. It would be better to state or highlight this clearly.

**Have the authors made all data and (if applicable) computational code underlying the findings in their manuscript fully available?**

Reviewer #1: Yes

Reviewer #2: Yes

Reviewer #3: None

PLOS authors have the option to publish the peer review history of their article (what does this mean? ). If published, this will include your full peer review and any attached files.

**Do you want your identity to be public for this peer review?** For information about this choice, including consent withdrawal, please see our Privacy Policy .

Reviewer #1: No

Reviewer #2: No

Reviewer #3: No

**Figure resubmission:**
---

## [Decision Letter · Decision Letter 1]

21 Jul 2025

Dear Dr Brown,

We are pleased to inform you that your manuscript 'A neural network model enables worm tracking in challenging conditions and increases signal-to-noise ratio in phenotypic screens' has been provisionally accepted for publication in PLOS Computational Biology.

Best regards,

Adriana San Miguel

Academic Editor

PLOS Computational Biology

Tobias Bollenbach

Section Editor

PLOS Computational Biology

Dear authors,

Your revised manuscript has now been assessed by three reviewers. I am pleased to inform you that your manuscript has been accepted for publication. Please look carefully at the comments by reviewer #2 to include the needed clarification.

Reviewer's Responses to Questions

**Comments to the Authors:**

Reviewer #1: The authors have answered all my queries and I now recommend publication.

Reviewer #2: I understand the difficulty in providing a more comprehensive method comparison with other approaches. I also appreciate a slightly more thorough estimation of tracking accuracy. My only comment is to the text in lines 180-181: "we manually corrected tracks for 3 and 15 worm videos" - what do these 2 numbers refer to? How many video frames does this represent? Clarifying this would allow to better understand the extent of tracking validation.

Reviewer #3: The authors have addressed all the comments and incorporated the suggestions. I, therefore, recommend the publication of this article.

**Have the authors made all data and (if applicable) computational code underlying the findings in their manuscript fully available?**

Reviewer #1: Yes

Reviewer #2: Yes

Reviewer #3: Yes

PLOS authors have the option to publish the peer review history of their article (what does this mean? ). If published, this will include your full peer review and any attached files.

**Do you want your identity to be public for this peer review?** For information about this choice, including consent withdrawal, please see our Privacy Policy .

Reviewer #1: No

Reviewer #2: **Yes: ** Katarzyna Bozek

Reviewer #3: No

---

## [Editor Report · Acceptance letter]

PCOMPBIOL-D-25-00022R1

A neural network model enables worm tracking in challenging conditions and increases signal-to-noise ratio in phenotypic screens

Dear Dr Brown,

I am pleased to inform you that your manuscript has been formally accepted for publication in PLOS Computational Biology. Your manuscript is now with our production department and you will be notified of the publication date in due course.

With kind regards,

Judit Kozma
